# Ostwald Ripening and Antibacterial Activity of Silver Nanoparticles Capped by Anti-Inflammatory Ligands

**DOI:** 10.3390/nano13030428

**Published:** 2023-01-20

**Authors:** Romina Alarcon, Mariana Walter, Maritza Paez, Manuel Ignacio Azócar

**Affiliations:** Departamento de Quimica de los Materiales, Facultad de Quimica y Biologia, & SMAT-C, Universidad de Santiago de Chile, Av. Bernardo O’Higgins 3363, Estacion Central, Santiago 8990000, Chile

**Keywords:** silver, nanoparticles, antibacterial

## Abstract

Silver nanoparticles (AgNPs) have been extensively studied during recent decades as antimicrobial agents. However, their stability and antibacterial activity over time have yet to be sufficiently studied. In this work, AgNPs were coated with different stabilizers (naproxen and diclofenac and 5-chlorosalicylic acid) in different concentrations. The suspensions of nanostructures were characterized by transmission electron microscopy, UV–Vis and FT-IR spectroscopic techniques. The antibacterial activity as a function of time was determined through microbiological studies against *Staphylococcus aureus*. The AgNPs show differences in stabilities when changing the coating agent and its concentration. This fact could be a consequence of the difference in the nature of the interaction between the stabilizer and the surface of the NPs, which were evaluated by FT-IR spectroscopy. In addition, an increase in the size of the nanoparticles was observed after 30 days, which could be related to an Ostwald maturation phenomenon. This result raises new questions about the role that stabilizers play on the surface of NPs, promoting size change in NPs. It is highly probable that the stabilizer functions as a growth controller of the NPs, thus determining an effect on their biological properties. Finally, the antibacterial activity was evaluated over time against the bacterium Staphylococcus aureus. The results showed that the protective or stabilizing agents can play an important role in the antibacterial capacity, the control of the size of the AgNPs and additionally in the stability over time.

## 1. Introduction

In the last decade, silver has been more widely studied as nanoparticles due to its antimicrobial ability against a wide range of bacteria, fungi and even viruses such as SARS-CoV-2 [1,2,3,4]. Due to these qualities, it is the most studied nanomaterial for applications in medicine [5].

Features such as high antimicrobial efficiency and low resistance of microorganisms make nanoparticles well suited for products in various biocidal applications, including wound healing, food packaging, textiles, medical devices, air purification, water disinfection and animal husbandry [5]. Additionally, bacteria have become increasingly resistant to antibiotic treatments, creating a problem for human health. Therefore, there is a great incentive for and interest in developing new broad-spectrum antibacterial agents. For example, the formation of biofilms on the surface of medical devices can lead to chronic infections due to the resistance of bacteria to antibiotics [5].

One of the advantages of nanoparticles is that size and shape of these nanomaterials can be controlled through various experimental conditions [6,7]. Additionally, the nanostructures’ stability over time can be ensured by modifying their surfaces with functionalizing or protective agents [8,9].

The mechanisms associated with antibacterial action can be classified in two ways: the first one is the generation of reactive oxygen species (ROS) that has been previously related to the antimicrobial properties of silver compounds [10]. The most important ROS studied in the context of AgNPs are singlet oxygen, hydrogen peroxide, superoxide radical anion and hydroxyl radicals [10,11,12,13,14]. These reactive oxygen species can cause severe damage to DNA, RNA, lipids and proteins [14,15]. The second mechanism is related to surface oxidation of the metal nanoparticle from Ag(O) to Ag(I) as oxide [16,17]. The coating on the NP may passivate the surface and reduce the subsequent oxidation. However, silver oxide over the surface is necessary to dissolve and release bactericidal Ag^+^ in water. Accordingly, an aerobic medium is necessary to activate the biocidal properties of silver [18].

AgNPs can act as a generating source of Ag^+^ ions [19], therefore, control of its release can optimize and control the antibacterial activity, as well as reduce the cytotoxic effects [16,20]. This phenomenon could be controlled by a stabilizing agent on the metal surface [21,22].

In summary, the antibacterial activity of AgNPs will depend on not only factors such as size and shape, but also the availability of oxygen and the type of coating on their surface, among others [16,23,24,25]. Previous research on this topic suggests that the antibacterial activity of nanoparticles or compounds would be influenced by the nature of the atoms that are attached to the metal [16,20,21,23,26].

It has been shown that the silver–oxygen interaction is weaker and has greater antibacterial activity than in compounds with silver–sulfide or silver–nitrogen interactions. These binding differences have been related to the nature of the ionic bond between silver and the bound ligand, and their magnitude directly influences the release of silver ions into the medium [21,26].

In this sense, the affinity between silver ions and the carboxylate group has shown great versatility in their ionic union [27,28]. The examination of infrared spectroscopy has shown that the analysis of the carboxylate group provides relevant information on the nature of the “carboxylate–metal” bond. In this work, it is expected that the technique will allow the evaluation of the type of interaction of the AgNP surfaces with the carboxylate group (COO-) present in the different capping molecule [22,27,28,29]. The information obtained about the “molecular structure–composition” behavior would allow obtaining a variable to be correlated with the release of the Ag^+^ ion and, consequently, with the stability of the nanostructures over time.

In previous research, it has been possible to determine the affinity of carboxylate groups and silver ions in anti-inflammatory molecules [30]. This type of compound could not only stabilize the nanoparticles in variable ways, but also provide an additional property in potential medical uses [31]. While other silver formulations have clinical limitations due to negative effects on eukaryotic cell lines, existing data on silver carboxylate show the ability of controlled and reduced dosing cytotoxicity potential [32].

In this work, we study capping molecules having COO^−^ groups in their structure such as naproxen and diclofenac and 5-cholorosalicylic acid as protective agents. The role of the coating on the silver nanoparticles is analyzed against its antibacterial activity and stability over time.

## 2. Materials and Methods

### 2.1. Reagents and Broths

All chemical reagents were purchased from Merck or Sigma-Aldrich Co and used as received: naproxen (99%), diclofenac sodium (99%), NaBH_4_ (97%) and 5-cholorosalicylic acid (98%) and high purity AgNO_3_ (≥99.0%). Milli-Q water was used to prepare all the samples. Mueller–Hinton broth (MHB) was purchased from Difco.

### 2.2. Synthesis of Silver Nanoparticles (AgNPs)

AgNPs were synthesized by chemical reduction of AgNO_3_ in aqueous solution (5 mL, 1 × 10^−3^ M) with NaBH_4_ (5 mL, 3.0 × 10^−3^ M), in the presence of 5 mL of the stabilizers at different concentrations ([1]: 3 × 10^−3^ M; [2]: 1 × 10^−3^ M y [3]: 1 × 10^−4^ M): naproxen (Nap), diclofenac (Dic) and 5-cholorosalicylic acid (5-Cl). As a control, a suspension of AgNPs-Bare, without stabilization, was prepared. The synthesis was carried out by mixing the stabilizer solutions with NaBH_4_ with vigorous stirring at room temperature. Finally, the silver solution was added dropwise. All AgNP suspensions were stored in amber vials and kept in the dark.

### 2.3. Characterization of Silver Nanoparticles

The stability over time of the AgNP suspensions was studied in a Shimadzu 1800 UV spectrophotometer and the spectra were recorded from 350–700 nm. The samples were measured in 3 mL quartz cuvettes, where 0.5 mL of AgNP suspension was added and mixed with 2 mL of Milli-Q water. The size and shape of the AgNPs were analyzed with a HITACHI HT7700 transmission electron microscope. Samples were prepared on 400 mesh copper grids and dried at room temperature (25 °C). The coatings of the nanoparticles were studied by IR spectrophotometry, with Shimadzu IR Tracer-100 equipment. The spectra were recorded in KBr pellets and in a range of 4000 to 400 cm^−1^. The samples were dried before being prepared.

### 2.4. Antibacterial Activity

The antibacterial activity was studied by determining the values of minimal inhibitory concentration (MIC) for a percentage of inhibition ≥ 97% against *Staphylococcus aureus* (BAA-977). With the MIC value obtained by serial dilutions for the different suspensions, they were repeated with a selected concentration for 7 weeks, and following the protocols recommended by the CLSI (www.clsi.org, accessed on 13 January 2023).

In sterile 96-well plates, the different volumes of suspension were added, 2 µL of cultured bacterial inoculum and diluted to 0.5 McFarland (5 × 10^5^ CFU/mL) and with MH broth up to 200 µL for the determination of the MIC and for the studies of stability over time. Additionally, sterility controls were performed: growth (MHB + bacteria), solvent (MHB + solvent + inoculum) and Mueller–Hinton liquid culture medium (MHB).

Subsequently, the plates were incubated for 24 h at 37 °C and the absorbance at 600 nm was determined in an ELISA reader (Multiskan GO Model, Thermos Labsystems, Waltham, MA, USA). Bacterial growth was recorded every hour, and the absorption of AgNPs and Muller–Hinton broth (MHB) was measured. The growth control was determined from the mixture of MHB with inoculum of bacteria. All MIC experiments were performed in technical triplicate (3 wells per sample) and analyzed as the average of the 3 measurements.

## 3. Results

In order to evaluate the stability in solution and in ambient light, the samples were monitored by UV–Vis spectroscopy. The absorption spectra of each AgNP system were measured every 7 days (5 weeks). Figure 1 shows the plasmon bands of AgNP-5-Cl [1-2-3], AgNP-Dic [1-2-3], AgNP-Nap [1-2-3] and AgNP-Bare after 1, 7, 14, 21 and 28 days.

All UV–Vis spectra of the bare NP suspensions show a rapid decrease in absorbance with time and the formation of a precipitate. However, this behavior was reduced when the samples were stored in the dark and in an oxygen-free atmosphere. Previous studies show that this type of response in the UV–Vis spectra of AgNPs could be related to an increase in the size of the NPs and/or their aggregation [33,34,35].

In contrast, the coated AgNPs show a greater stability of their plasmonic band when the suspensions age, because the intensity of absorbance and the position in the spectrum vary less compared to the samples without coating.

The case of AgNPs stabilized with naproxen and diclofenac shows a decrease in absorbance, and the formation of second bands over time, which could be associated with a long exposure to the aqueous media, allowing aggregation, or growth, of the AgNP samples [36]. Therefore, the nature and concentration of the protective agents may play a role in the stability, or the formation of aggregates, of AgNPs; the non-stability of the uncoated silver nanoparticles is evident in Figure 1a.

As potential antibacterial agents, it is important that nanoparticle suspensions can be stable over time.

To understand the effect of time on nanoparticle suspensions, freshly prepared samples and samples after 30 days were analyzed. Figure 2 shows the appearance of the sample without a coating agent over time.

The size distribution analysis shows a decrease in size after 30 days. Initially, the nanoparticles have an average of 28.7 nm, but over time, they decrease in size to an average of 11.0 nm. The release of silver ions from the surface of AgNPs, deliberately oxidized by an oxidizing agent, is described as following the sequence of reactions below [23]:4 Ag^0^ (NPs) + O_2_ → 2 Ag_2_O(1)
2 Ag_2_O + 4 H+ → 4 Ag^+^ + 2 H_2_O(2)

This phenomenon could explain the decrease in pH as the nanoparticles age, as recorded in Table 1.

Considering the presence of coating or stabilizing agents, we hypothesized that the nanoparticles with naproxen and diclofenac could reduce the dissolution effect over time.

However, surprisingly, the TEM analyses show a noticeable increase in the size of the nanoparticles. Figure 3 shows the effect of the coating agent concentration and the size comparison after 30 days. For AgNP-Dic (1), the size differences are clear, where the initial sample has a size of 3.5 nm and later reaches an average size of 10.2 nm.

In the case of AgNP-Dic (2), this phenomenon of size increase was not observed, however, in AgNP-Dic (3) an increase in the size distribution is observed, reaching values of up to 60 nm and also sizes close to 2 nm.

In the case of the naproxen-coated agent, the increase in sizes after 30 days was more evident, being observed in the three concentrations of the carboxylic ligand used(See Figure 4). This phenomenon can be strongly linked to the Ostwald ripening due to the disappearance of the smallest particles and the appearance of particles on average greater than 15 nm in all cases and reaching up to 40 nm for Nap-2 and Nap-3.

Ostwald ripening is a phenomenon characterized by a growth mechanism and is caused by the solubility of the nanoparticles and depends on their size. This phenomenon has been described using the Gibbs–Thomson relationship, according to the following equation:$$C_{r}=C_{b}\exp\;\frac{2yv}{{}_{r}k_{B}T}$$
where the solubility of nanoparticles depends on their size and, according to the Gibbs–Thomson relationship, a spherical particle has an extra chemical potential ∆*µ* = 2*γ*v/r.

Then, Cr is expressed as a function of r, where v is the molar volume of the bulk crystal and Cb is the concentration of the bulk solution [37].

Due to the high surface energy and solubility of nanoparticles in solution, they dissolve, allowing larger particles to grow even larger. In this stage of growth of the metallic nanoparticles, the Ostwald ripening phenomenon occurs, which results in changes in size [38].

Various study strategies have been developed for this phenomenon, such as blast nucleation and surface stabilization with coating agents, to inhibit Ostwald ripening [39,40].

In the case of the ligands used, different proposals have tried to explain the Ostwald ripening with the nature of the ligand–surface interaction. Mainly two effects can be considered: the strength of the ligand–NP surface interaction, and the labile nature of the ligands either in free form or in the form of complexes and clusters that they form with the NP constituents. When evaluating the three concentrations, it is observed that a decrease in the stabilizing agent caused, in all cases, an increase in the size distribution. These ligands, depending on the strength of the ligand–NP surface interaction, can desorb from the surface, exposing bare NP surfaces when the ripening process is underway. If such bare surfaces come together, the NPs can merge, leading to their uncontrolled growth instead of remaining at their initial size. On the other hand, if excess ligands are used, they will be present in the surrounding solvent environment of the NPs. These free ligands will (a) make it difficult for the ligand to desorb (because the solvent is already saturated by excess free ligands) and (b) immediately replace the ligand even if it is desorbed, thus preventing NP growth [39].

In this sense, the interaction of the carboxylate groups and the concentration present in the medium may play an important role and influence the ripening phenomenon.

To understand the interaction between the coating agents and the surface of the nanoparticles, an infrared spectroscopic study of the carboxylate groups was carried out.

The infrared spectroscopy technique has allowed us to study the signals of the carboxylate groups, which show two types of characteristic signals: symmetrical (vs.) and asymmetrical (vas) stretching. From these analyses and the energy difference between the two types of stretching (Table 2, ∆v = vas (COO-)—vs. (COO-), it has been possible to determine the nature of the metal–carboxylate bond [27,28,31,41].

The interaction of the carboxylate group of the stabilizer in IR spectroscopy indicates that as the concentration of stabilizer decreases in AgNP-Dic, the interaction of the stabilizer with the surface of the NPs becomes more covalent, and therefore more stable [41], being consistent with a greater ripening effect over time. Similarly, and with respect to the ∆v of the carboxylate group in AgNP-Nap, it indicates a more covalent interaction between the stabilizer and the surface of the NPs at lower concentrations and therefore a more controlled growth.

Finally, for AgNP-5-Cl, the infrared spectra show values greater than 240 cm^−1^, which have been described for covalent silver–carboxylate interactions. These results for the three samples analyzed can account for the greater stability of the absorption band in the UV–Vis spectra and less Ostwald ripening effect.

Therefore, the ionic character of this ligand–metal interaction could generate an easy and/or rapid dissolution of the NPs that could account for the increase in the size of AgNPs after 30 days [36,42,43].

### Antibacterial Activity

The % inhibition of AgNPs against the microorganism *Staphylococcus aureus* was tested at 20 µg/mL, where the stabilized AgNP maintained higher inhibition values over time, demonstrating the effect of the surface coatings on nanostructures.

AgNP-Dic and AgNP-Nap showed lower MIC variations over time with respect to AgNP-Bare as shown in Figure 5. These characteristics could be associated with an silver–carboxylate interaction for naproxen-stabilized nanoparticles and also a molecular size effect for diclofenac. This interaction of the coating with the metal surface could be related to a slower and more prolonged release of silver ions from the surface and a continuous loss of AgNP-Bare activity associated with oxidation–agglomeration phenomena and subsequent precipitation.

The results in Figure 2 show, for all coated AgNPs, low variations in the absorbance of the plasmon bands over time [16]. This could corroborate that the dissolution mechanism of coated AgNPs is slow and prolonged over time, even when the dissolution or release of Ag^+^ ions from the AgNPs in the presence of oxygen is stimulated [20].

Several authors propose the study of the dissolution, aggregation and sedimentation of AgNPs through UV–Vis spectroscopy. A decrease in absorbance in the spectrum may indicate the aggregation or dissolution of AgNPs [35,42,44]. Other authors mention the appearance of second bands from 500–600 nm related to the aggregation of NPs [36]. However, these phenomena appear after several days of exposure to light and mainly to oxygen.

In the case of AgNP-5-Cl, it shows a loss of antibacterial activity after 7 weeks, similar to AgNP-Bare. Although the 5-Cl-coated sample is stable according to UV–Vis and FT-IR spectroscopy, it may lose antibacterial activity like uncoated samples (AgNP-Bare). Therefore, the nature of the stabilizer can also influence the stability and antibacterial activity, as well as the size changes of the silver nanoparticles.

## 4. Conclusions

The use of diclofenac, naproxen and 5-chlorosalicylic acid as AgNP protection agents allows control of the growth of nanoparticles and, eventually, agglomeration phenomena. When relating the results found with the possible medical applications of AgNPs, it is important to consider the use of coating agents for stability and long-term antibacterial activity. A substantial reduction in the potential cytotoxic effects of nanostructures can be achieved, mainly due to the more controlled release of silver ions.

## Figures and Tables

**Figure 1 nanomaterials-13-00428-f001:**
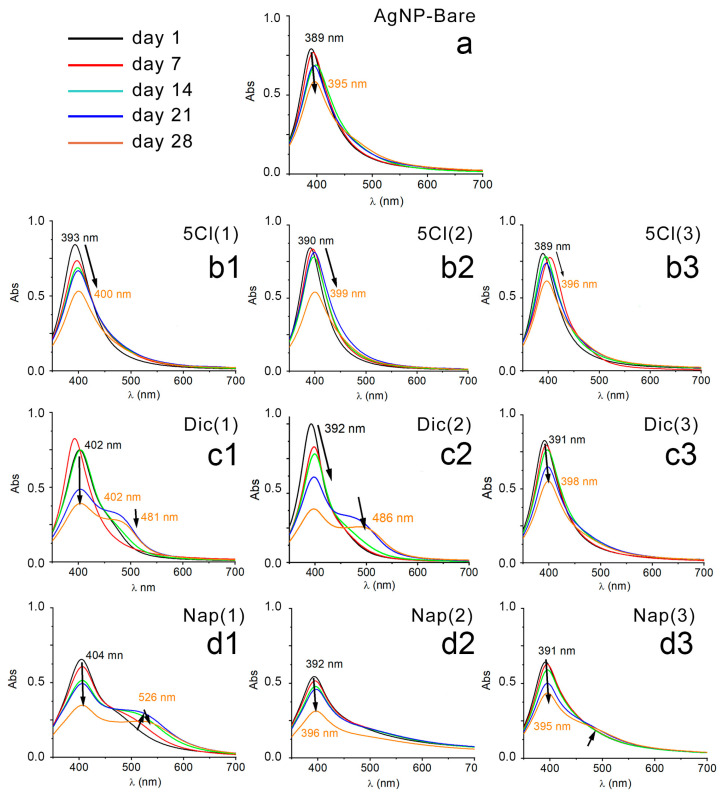
UV–Vis spectrum: (**a**) AgNP-Bare, (**b1**) AgNP-5-Cl (1), (**b2**) AgNP-5-Cl (2), (**b3**) AgNP-5-Cl (3), (**c1**) AgNP-Dic (1), (**c2**) AgNP-Dic (2), (**c3**) AgNP-Dic (3), (**d1**) AgNP-Nap (1), (**d2**) AgNP-Nap (2), (**d3**) AgNP-Nap (3), every 7 days and up to 28 days.

**Figure 2 nanomaterials-13-00428-f002:**
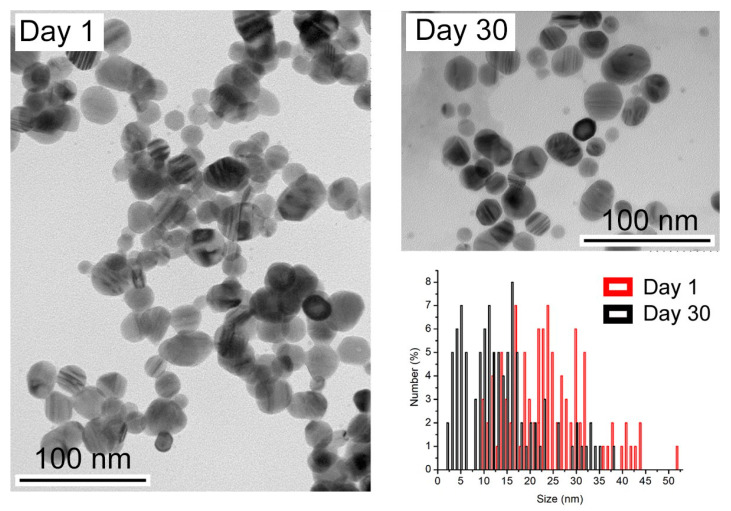
TEM images of the synthesized AgNPs and after 30 days. Size distribution of AgNPs calculated from TEM images.

**Figure 3 nanomaterials-13-00428-f003:**
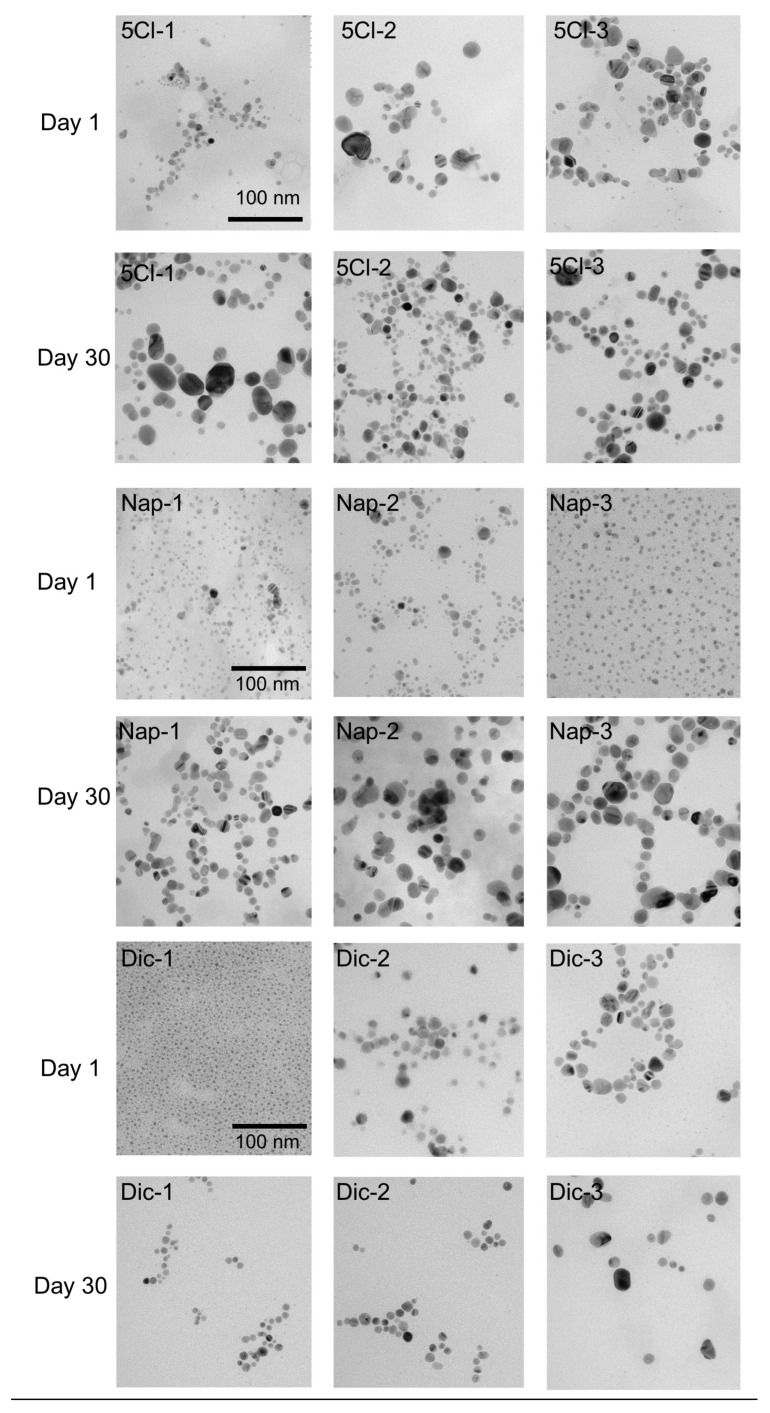
TEM images of the synthesized AgNPs and after 30 days.

**Figure 4 nanomaterials-13-00428-f004:**
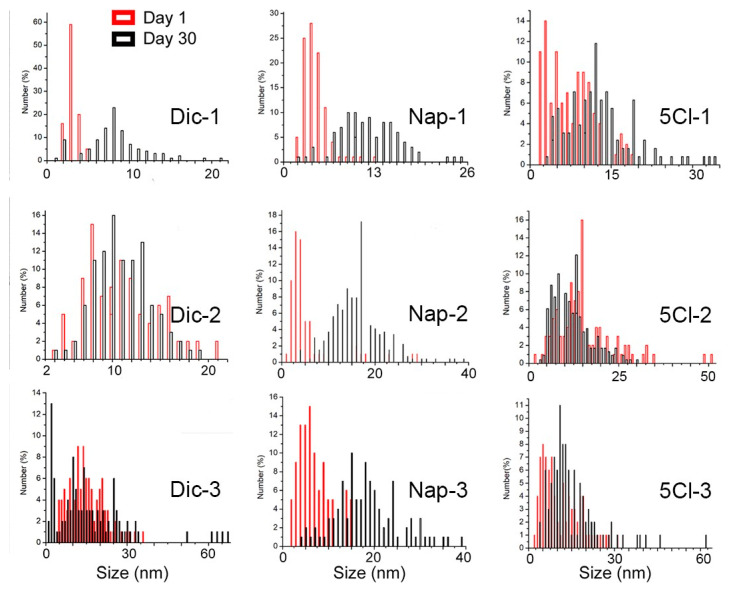
Size distribution of AgNPs calculated from TEM images.

**Figure 5 nanomaterials-13-00428-f005:**
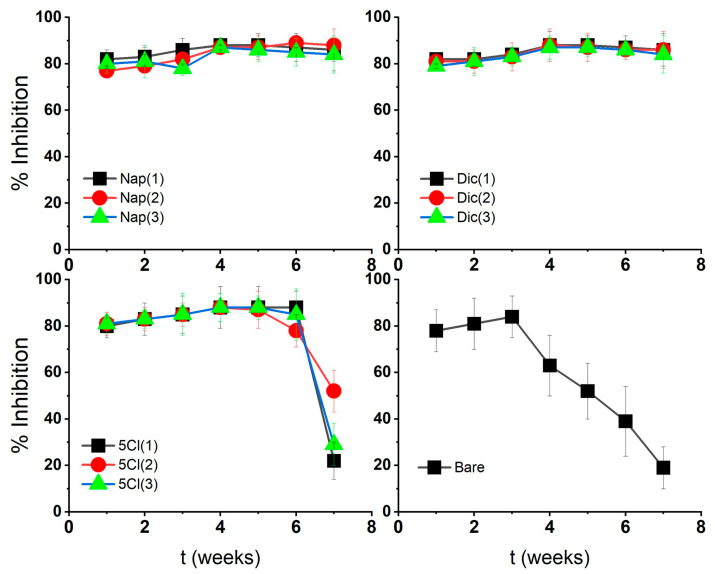
Antibacterial activity over time of silver nanoparticles against *Staphylococcus aureus*.

**Table 1 nanomaterials-13-00428-t001:** pH of NP suspensions: day 1 and day 30.

AgNP-Dic	AgNP-Nap	AgNP-5-Cl	AgNP
Day→(1)→(2)	(3)→(1)→(2)	(3)→(1)→(2)	(3)→Bare
1→8.92→8.63	8.59→8.36→8.62	8.61→9.64→8.86	8.75→8.67
30→7.73→7.09	7.47→7.18→7.42	7.59→7.79→7.94	7.90→7.94

**Table 2 nanomaterials-13-00428-t002:** Infrared frequency values of the symmetric and asymmetric stretching of the carboxylate groups.

*ν* COO- (cm^−1^) AgNP-Dic		*ν*as COO- (cm^−1^)	∆(*ν*a *− ν*)	[M] mol/L
1	1397	1575	178	3 × 10^−3^
2	1386	1575	189	1 × 10^−3^
3	1378	1614	236	1 × 10^−4^
Diclofenac	1401	1575	174	
AgNP-Nap				
1	1408	1604	196	3 × 10^−3^
2	1380	1604	224	1 × 10^−3^
3	1376	1613	237	1 × 10^−4^
Naproxen	1394	1584	190	
AgNP-5-Cl				
1	1384	1627	243	3 × 10^−3^
2	1383	1630	247	1 × 10^−3^
3	1383	1627	244	1 × 10^−4^
	1363	1527	164	

## Data Availability

Not applicable.

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
