# Peer review of "Ostwald Ripening and Antibacterial Activity of Silver Nanoparticles Capped by Anti-Inflammatory Ligands"

_nanomaterials, 2023, doi:10.3390/nano13030428_

Round 1

Reviewer 1 Report

The authors presented an interesting manuscript in which they showed that the use of diclofenac, naproxen, and 5-chlorosalicylic acid as AgNPs protection agents allows to control of the growth of nanoparticles and agglomeration phenomena. 

Please refer to the comments below:

1. The authors should additionally analyze the stability of the obtained hydrocolloids of silver nanoparticles (zeta potential analysis)

2. The manuscript contains a lot of punctuation errors: line 42 - remove the space before the period, add a space after the period. Similarly in lines 42, 50, and many others

Author Response

I really appreciate all your comments. We have corrected the errors mentioned in the different lines of the manuscript.

For this research we included different variables to explain the activity and stability of the different silver nanoparticles.

Unfortunately we have not included an analysis of zeta potential, but we consider in our future works to give more importance to this technique, and thus improve our size analysis and the proposal of the Ostwals ripening phenomenon.

Reviewer 2 Report

  This work focus on Ostwald ripening and antibacterial activity of silver nanoparticles capped by anti-inflammatory ligands. This research has strong theoretical significance and practical value. The authors should address the following issues.

(1)  In addition to the Ostwald Ripening, the increase in nanoparticles sizes can be caused by cold-melting effect between nanoparticles. The authors should explain the influence of cold-melting effect.

(2)  It lacks error bars in Figure 5 to support author’s discoveries.

(3)  The authors should explain the difference between three anti-inflammatory ligands.

(4)  Figure 3 should be adjusted for better presentations.

(5) The authors are advised to quote the following articles.

 Silver nanocubes monolayers as a SERS substrate for quantitative analysis, DOI: 10.1016/j.cclet.2020.10.021”, Chinese Chemical Letters 32 (2021) 1497–1501.

“Ag-Activated Metal-Organic Framework with Peroxidase-like Activity Synergistic Ag+ Release for Safe Bacterial Eradication and Wound Healing”, Nanomaterials, 2022, 12, 4058. https://doi.org/10.3390/nano12224058.

Author Response

1. In addition to the Ostwald Ripening, the increase in nanoparticles sizes can be caused by cold-melting effect between nanoparticles. The authors should explain the influence of cold-melting effect.

R1. Thank you very much for your comments and suggestions. We are very surprised by an additional effect to explain the size change of nanoparticles from our research: influence of cold-melting effect. However, we are unaware of this effect and we would like to include this new point of view in our work. Can you suggest publications related to this phenomenon? We really appreciate your suggestions.

2. There are no error bars in Figure 5 to support the author's findings.

R2. Thank you very much for your comment. We have corrected the graphs and now they have error bars on the Y axis.

(3) Authors should explain the difference between the three anti-inflammatory ligands.

R3. In previous research, it has been possible to determine the affinity of carboxylate groups and silver ions in anti-inflammatory molecules. This type of compound could not only stabilize the nanoparticles in variable ways, but also provide an additional property in potential medical uses. While other silver formulations have clinical limitations due to negative effects on eukaryotic cell lines, existing data on silver carboxylate shows the ability of controlled and reduced dosing cytotoxicity potential.

REF: METTE, Makena et al. Silver Carboxylate as an Antibiotic-Independent Antimicrobial: A Review of Current Formulations, in vitro Efficacy, and Clinical Relevance. Medical Research Archives, [S.l.], v. 10, n. 12, dec. 2022. ISSN 2375-1924.

REF2: Aldabaldetrecu, M.; Tamayo, L.; Alarcon, R.; Walter, M.; Salas-Huenuleo, E.; Kogan, M.J.; Guerrero, J.; Paez, M.; Azócar, M.I. Stability of Antibacterial Silver Carboxylate Complexes against Staphylococcus epidermidis and Their Cytotoxic Effects. Molecules 2018, 23, 1629. https://doi.org/10.3390/molecules23071629 

4. Figure 3 should be adjusted for better presentations.

R4. Thanks a lot for the suggestion. Figure 3 has been modified for better presentation.

5. Authors are encouraged to cite the following articles.

R5. Thank you very much for the recommendation. The 2 articles are very interesting and have been included in the manuscript. (refs 4 and 6)

Reviewer 3 Report

1. The abstract should be refined.

2. In the introduction section, the author should be introduced the effect of protective agents on the antibacterial activity of silver nanoparticle, and the reasons to choose the Naproxen, diclofenac, and cholorosalicylic acid as research object. In addition, the logic and language of the preface should be  further improved.

3. Some abbreviations and silver ions writing should be unified, for example, ROS and Ag+, mL, AgNPs, Uv-vis, etc.

4. In lines 82 and 83,  there are some typo in concentration, for example,  3 x10-3  M; [2]: 1 x10-3  M y [3]: 1 x10-4  M.

5. In Figure 1, the pictures label is missing.

6. The size increase, the anthor explain it caused by the Ostwald ripening phenomenon. However, the authors did not give a reason for the differences among the three protective agents.

7. In experimential section, the author claims that the MIC assay was following the protocol recommended by the CLSI. But in the result, there was not MIC results, why?

8. In the introduction section, some references should be cited and discussed the advantage of protective agent modification strategy compared with rapid preparation strategy, for example, 10.1038/s41392-022-00900-8; 10.1021/acs.jpclett.2c01737.

Author Response

1. The abstract should be refined.

R1. Thank you very much for the correction. The abstract has been refined and a new version has been included in the manuscript.

2. In the introduction section, the author should be introduced the effect of protective agents on the antibacterial activity of silver nanoparticle, and the reasons to choose the Naproxen, diclofenac, and cholorosalicylic acid as research object. In addition, the logic and language of the preface should be  further improved.

R2. Thanks a lot for the suggestion. The introduction has been improved and 3 new references have been included: 28, 29 and 30. References 28 and 29 describe our investigations with silver carboxylates, including the use of anti-inflammatory ligands. Reference 30 (Review) has recently been published in 2022 and describes the clinical importance of silver carboxylates. Ref 30: Silver as an Antibiotic-Independent Antimicrobial: Review of Current Formulations and Clinical Relevance. Surgical Infections. Nov 2022.769-780. http://doi.org/10.1089/sur.2022.229.

3. Some abbreviations and silver ions writing should be unified, for example, ROS and Ag+, mL, AgNPs, Uv-vis, etc.

R3. Thank you very much for the corrections.

All abbreviations have been checked and corrected in the manuscript.

4. In lines 82 and 83,  there are some typo in concentration, for example,  3 x10-3  M; [2]: 1 x10-3  M y [3]: 1 x10-4  M.

R4. Thank you very much for the corrections.  The manuscript has been revised and corrected.

5. In Figure 1, the pictures label is missing.

Thank you very much for the corrections. All labels have been included in the Figure 1.

6. The size increase, the anthor explain it caused by the Ostwald ripening phenomenon. However, the authors did not give a reason for the differences among the three protective agents.

Thank you very much for such an interesting and insightful observation. We have described in greater detail the effect of the coating and also improved the explanation and the comparison between the 3 ligands.

R6. In the case of the ligands used, different proposals have tried to explain the Ostwald Ripening with the nature of the ligand-surface interaction. Mainly 2 effects can be considered: The strength of the ligand-NP surface interaction, and The labile nature of the ligands either in free form or in the form of complexes and clusters that it forms with the NP constituents. When evaluating 3 concentrations, it is observed that a decrease in the stabilizing agent caused in all cases an increase in the size distribution. These ligands, depending on the strength of the ligand-NPs surface interaction, can desorb from the surface, exposing bare NPs surfaces when the Ripening process is underway. If such bare surfaces come together, the NPs can merge, leading to their uncontrolled growth instead of focusing on initial size. On the other hand, if excess ligands are used, they will be present in the surrounding solvent environment of the NP. These free ligands will (a) make it difficult for the ligand to desorb (because the solvent is already saturated by excess free ligands) and (b) will immediately replace the ligand even if it is desorbed, thus preventing NP growth.

7. In experimential section, the author claims that the MIC assay was following the protocol recommended by the CLSI. But in the result, there was not MIC results, why?

R7. That is a very good observation. In the first stage, we evaluated the MIC of each system. But this did not allow a good comparison of the performance of each system. For this reason, we set a value to reach at least 80% inhibition of all samples (10) and make a comparison over time. With this method we were able to graph and show the differences more clearly.

8. In the introduction section, some references should be cited and discussed the advantage of protective agent modification strategy compared with rapid preparation strategy, for example, 10.1038/s41392-022-00900-8; 10.1021/acs.jpclett.2c01737.

Thank you very much for your suggestions. We have cited both articles because of their importance, and because they are very current articles.

Round 2

Reviewer 1 Report

I accept the present form of the manuscript

Reviewer 3 Report

after careful revision, the manuscript can be published in the current form.